# Predation history has no effect on lateralized behavior in *Brachyrhaphis rhabdophora*

**Maren G. Callaway** [ORCID]**\***, **Erik S. Johnson, Jerald B. Johnson**

Department of Biology, Evolutionary Ecology Laboratories, BYU Life Science Museum, Brigham Young University, Provo, UT, United States of America

\* marencallaway@gmail.com

**Data Availability Statement:** Additional data has been uploaded as a supporting information file.

**Funding:** This work was funded by Brigham Young University College of Life Sciences, College

## Abstract

Evolutionary biologists have grown increasingly interested in laterality, a phenomenon where bilaterally symmetrical organisms show a side bias in some trait. Lateralized behavior is particularly interesting because it is not necessarily tied to morphological asymmetry. What causes lateralized behavior remains largely unknown, although previous research in fishes suggest that fish might favor one eye over another to view potential food sources, mates, and to assess predation risk. Here we test the hypothesis that a history of predation risk predicts lateralized behavior in the livebearing fish *Brachyrhaphis rhabdophora*. To do this, we used a detour assay to test for eye bias when a focal fish approached various stimuli (predator, potential mate, novel object, and empty tank control). Contrary to our predictions, we found no differences in lateralized behavior between fish from populations that co-occurred with fish predators relative to those that do not co-occur with predators. In fact, we found no evidence for behavioral lateralization at all in response to any of the stimuli. We explore several possible explanations for why lateralized behavior is absent in this species, especially considering a large body of work in other livebearing fishes that shows that lateralized behavior does occur.

## Introduction

Laterality is a common and well-documented phenomenon in many vertebrates [1]. Up until the 1941 discoveries documenting handedness in chimpanzees [2], laterality was largely thought to be unique to humans. Since then, the evolution of laterality has received widespread interest. Laterality is often expressed in terms of morphology, such as differing claw sizes in crabs [3] or the asymmetry of turtle shells [4]. However, it has also been documented in behavior, where individuals have lateralized tendencies or preferences [1]. This phenomenon is surprisingly common in nature—research has documented lateralized behavior in a variety of organisms, including domestic chicks [5], dogs [6], toads [7], and even an extinct Paleozoic reptile [8]. Laterality is found in a wide range of behaviors; for example, chimps display preferential hand use [2], sea turtles preferentially use a dominant flipper to swim [9], passerine birds display "footedness" when catching mealworms [10], and humans display a bias in head-turning during kissing [11]. Interestingly, several fish species prefer to use one eye over the other when viewing certain stimuli [12–14]. For instance, research has documented lateralized

Undergraduate Research Awards of $3,000 each to Maren Callaway and Erik S. Johnson.

**Competing interests:** The authors have declared that no competing interests exist.

behavior by means of eye bias in many species of livebearing Poeciliid fishes [15, 16]. Such widespread occurrence of lateralized behavior suggests that it could be a heritable, adaptive trait. This raises the question about what factors might result in the selection of this trait over time.

Predation is an environmental factor known to affect behavior in a variety of organisms (e.g., mule deer [17], water striders [18], and some species of Poeciliid fishes [19]). Of particular interest is how predation can affect lateralized behavior, especially in fishes. Mosquitofish males, for example, preferentially use one eye to evaluate mates and predators, but show no bias for other males or an empty tank [14]. Recent work in Bahamas mosquitofish shows that individuals from high predation environments were more strongly lateralized than those from environments absent predators [20]. These findings echo the results of earlier work on another livebearing fish species, *Brachyrhaphis episcopi*, which found that individuals from high-predation environments displayed stronger lateralized behavior than those from low-predation environments [21, 22]. However, it remains unclear how universal these findings are, and to what extent the presence or absence of predators in a population's environment affects behavioral lateralization. Fortunately, freshwater fishes are often found in populations with and without predators. Hence, we can readily evaluate this question by studying additional systems using a similar experimental framework.

We have identified a fish species in the wild where populations exist in both predator and predator-free environments: the livebearing fish *Brachyrhaphis rhabdophora*. Populations of this species have independently evolved in the presence or absence of fish predators, and this is replicated across multiple river systems [23]. Additionally, we know that predation influences a variety of other traits in this species and others in its genus, such as life history [23, 24], morphology [25], personality [26], and growth rates and age-specific mortality rates [27]. If lateralized behavior is an adaptation to predation, we predict that individual *B. rhabdophora* from predation environments will show a stronger, more consistent lateral eye bias when viewing predators than fish from non-predation environments.

In this study, we address two questions. First, does *B. rhabdophora* show lateralized behavior in response to predators, as well as to a variety of other stimuli? Second, is there a difference in lateralized behavior between fish from a predator environment and fish from a predator-free environment? In other words, does predation history affect lateralized behavior? To address these questions, we tested for eye preference when an individual approaches predators, potential mates, and a novel object. While predation history differs between these populations, the mating context is consistent. Hence, we predicted that males would respond similarly to potential mates and to a novel object, and that if difference in lateralized behavior towards these stimuli exists, it would most likely be ascribed to differing predation histories of the two populations.

## Methods

### Study system

We used the livebearing fish species *Brachyrhaphis rhabdophora*, a species endemic to Costa Rica [28], to test the hypothesis that behavioral laterality would differ among populations from different predation environments. Like most other species in the family Poeciliidae, *B. rhabdophora* is morphologically symmetrical—the left and right sides are effectively mirror images of each other. We collected *B. rhabdophora* from two streams in Guanacaste Province: Rio Javilla (predator population) and Quebrada Grande (predator-free population), in June 2018 and April 2019, respectively (see Fig 1). The close geographic proximities (and therefore similar micro-climates) of these two streams allow us to make relevant comparisons between the

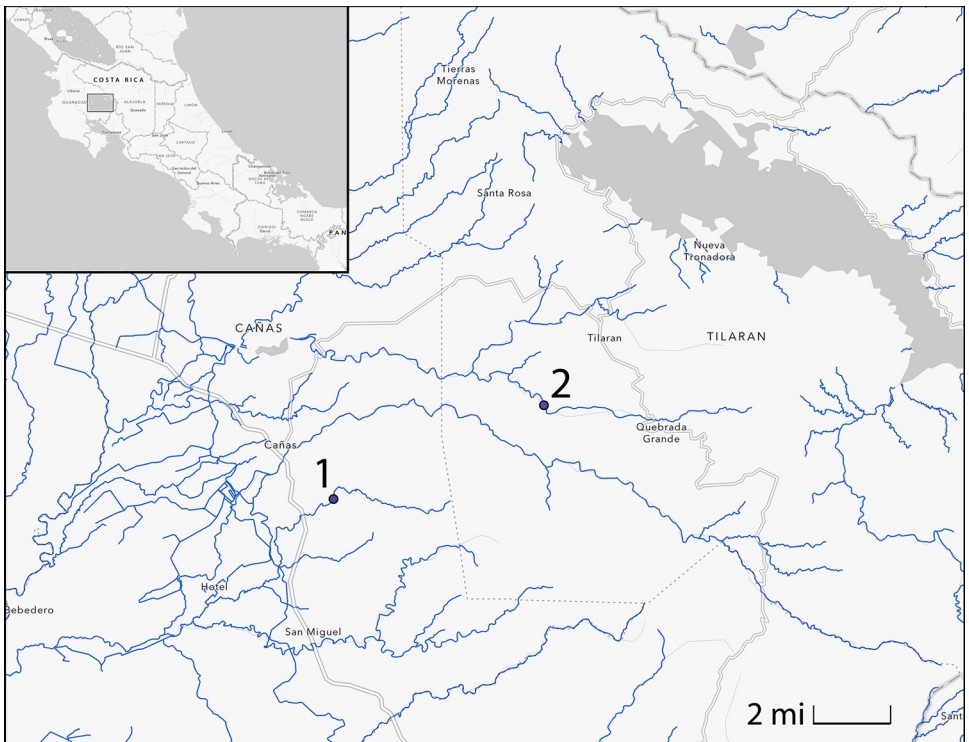

**Fig 1. Sampling sites in Northern Costa Rica.** 1) Rio Javilla (N10.40569, W-85.06948, elevation 112 m). 2) Quebrada Grande (N10.441976, W-84.98798, elevation 366 m). See Johnson & Belk [24] and Johnson [23] for additional details about these localities. This figure was generated in ArcGIS [29].

two populations of *B. rhabdophora* that inhabit them. Previous work has shown that these populations experience different mortality rates assumed to be due to differences in predation [27]. We collected 150 live fish from each locality and transported them to the breeding facility at Brigham Young University, where they were then treated for parasites and allowed to acclimate to laboratory conditions. The fish were held in 10-gallon tanks at 23C on a 12L:12D light cycle, being fed *ad libitum* twice daily. All aspects of this study were approved by Brigham Young University Animal Care and Use Committee via written consent (IACUC 18–0803).

## Experimental design

To answer our question about whether lateralized behavior differs between differing predation environments, we sought to identify whether or not *B. rhabdophora* showed a lateralized behavior in the eye used to approach a predator. We focused on male individuals, based on previous work in our laboratory in the livebearing fish *Xenophallus umbratilis* that showed that males with different intromittent organ morphs (left- or right-handed corkscrew terminus in the gonopodium) detoured in opposite directions from one another when approaching a predator [16]. In this study, we isolated 30 males in individual tanks: 15 from the predator population, and 15 from the predator-free population. This allowed us to test individuals with different stimuli over time. To determine lateral bias, we used a detour tank identical to Johnson et al. [16], which has been used to test for lateralized behavior in other species of the Poeciliidae family and is a common approach in behavioral research [16, 30]. The detour tank consists of an arena built as a raceway between two opposite sides of a tank (see Fig 2); the tank was lit uniformly from above with three full-spectrum bulbs illuminating the entire arena with no

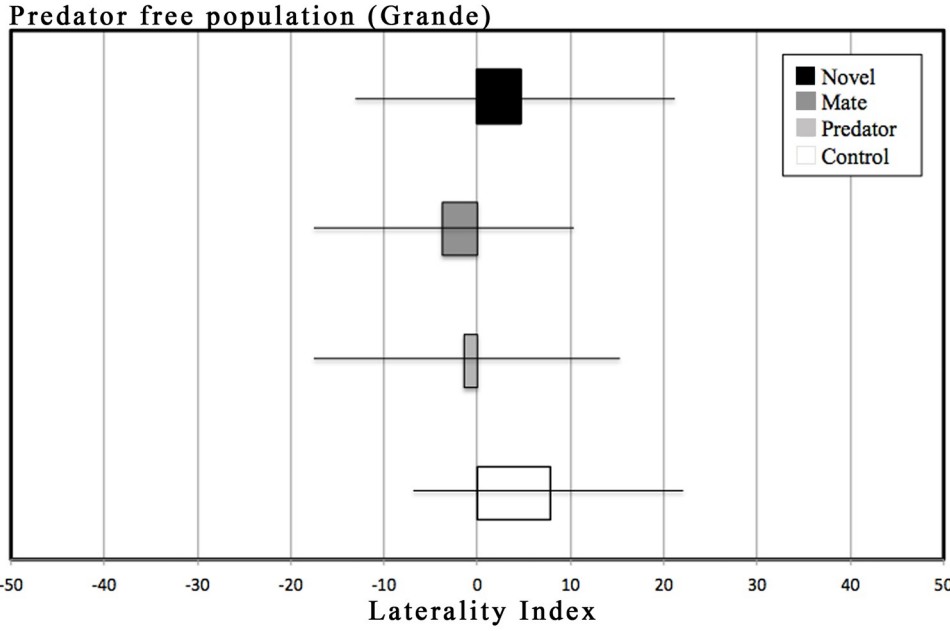

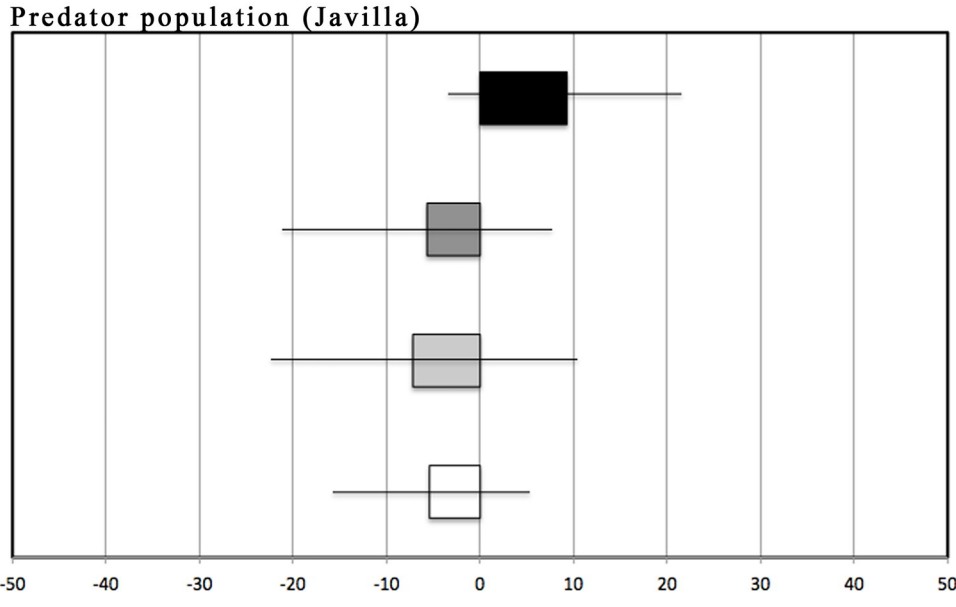

**Fig 2. Detour tank arena.** S = stimulus. See Johnson et al. [16].

shadows present. A male individual placed in the center of the raceway was allowed to swim back and forth from one end of the raceway to another. At each end, the male encountered a dividing barrier in the raceway and a visual stimulus that was partially obscured by a screen. To see the stimulus completely and clearly, the male had to choose to detour either left or right at the barrier. Teleost fishes cannot use both eyes to clearly focus on the same subject [31]; thus, detouring left or right reveals which eye the individual prefers to view the stimulus. Each time a male swam from one end to the other, he could make this detour choice. We did not set a minimum number of detour turns for individuals to be included in the analyses because we found no difference in our results when setting a minimum of five detours versus no such minimum. We tested for laterality using three stimuli and a control, predicting that *B*.

*rhabdophora* might have different lateral biases for different stimuli as has been shown in previous work on other livebearing fishes [13, 14, 16]. Our stimuli were: 1) predator (the fish *Parachromis dovii*, a known predator of *B. rhabdophora*) [27]; 2) potential mate (*B. rhabdophora* adult females); 3) novel object (a mixed-color block of LEGOs, something *B. rhabdophora* would have never previously encountered); and 4) control (an empty stimulus tank). For predators, we used two size-matched *P. dovii*. For potential mates, we isolated six females, randomly selecting two of them (one for each end of the detour tank) for each assay. For the novel object, a LEGO "Y" composed of orange, yellow, blue, white, and red blocks was placed at each end. This was an object that would be totally unfamiliar to the males, yet still potentially attracting their interest. Each individual participated in four trials total, one for each of the four stimuli. Fish from each population were first assayed with the control, followed by either the mate or the novel object, and with the predator as the final stimulus. Between trials, individuals were given at least three days of rest, which we deemed sufficient to allow fish to swim normally at a baseline rate. Following the trials, all individuals were returned to the breeding colony and made available for use in future research.

We tested all 30 individuals for lateral bias for each of the four stimuli. Given both the sample size of similar studies done with other species [14, 32] and the animal care objective to avoid unnecessary live animal trials, we used a sample size of 30. In the assays, a male was allowed to traverse the tank freely for 20 minutes. In order to ascribe any detour behavior solely to the stimulus, the tank was kept in a soundproof room and we remotely observed the trial using a camera mounted over the tank. During the assay, we scored each time a fish detoured left or right. After 20 minutes, we removed the male and filtered the water for 10 minutes before the next fish was assayed. With this design, we could determine if *B. rhabdophora* shows a lateral bias in detour behavior between stimuli by observing left and right detours, and then determined if that lateral bias differed between the two populations.

## Statistical analyses

We used the statistical approach of Torres-Dowdall et al. [32] to determine if *B. rhabdophora* show a lateral bias. This consisted of calculating a laterality index (LI) using the following equation:

$$\text{LI} = \frac{Detour\ Right - Detour\ Left}{Detour\ Right + Detour\ Left} * 100$$

LI measures the degree to which an individual shows a bias in left or right detours. An LI value of 0 would mean the individual shows no lateral bias. A positive LI value reflects a right-handed detour bias (using the left eye to view stimuli) while a negative LI value reflects a left-handed detour bias (using the right eye to view stimuli). To determine whether there was a bias toward each visual stimulus, we calculated a mean laterality score for each stimulus treatment for both the predator and predator-free populations (eight tests total). We used a two-tailed, one sample t-test to determine if the LI for each stimulus treatment significantly departed from 0. Additionally, we calculated a mean laterality score for each individual across the four stimulus treatments, and used a two-tailed, one sample t-test to determine if the LI for each individual significantly departed from 0. We also ran a mixed-model, repeated-measures ANOVA, treating population (predator versus non-predator) and treatment (four levels described above) as independent variables, and individual as a random variable (following [21]).

## Results

*Brachyrhaphis rhabdophora* males do not show lateralized behavior in detour behavior, regardless of stimuli or whether they come from a population without or without a history of co-occurring with fish predators. For each stimulus test we ran for both predator- and predator-free populations, we found that LI values did not differ significantly from 0 (see Fig 3), indicating that individuals detoured at random when approaching different stimuli. Interestingly, no fish in our trials showed strong individual laterality, defined as an individual LI score exceeding 0.7. Our repeated measures ANOVA also revealed no difference in behavioral laterality between predator environments (p = 0.64), or among the four experimental stimulus treatment (p = 0.38). All analyses point to an absence of lateralized behavior in this species.

## Discussion

Despite numerous studies that demonstrate lateral biases in detour behavior in fishes [12–16, 20, 22, 32], including biases in response to predators [16, 20] we found no evidence for lateralized behavior in *B. rhabdophora*. This was true for each of the four stimuli we evaluated, regardless of whether fish came from populations that had historically co-evolved with or without predators. In every assay we evaluated, fish detoured seemingly at random. Here, we consider two possible explanations.

### Absence of laterality

It is possible that *B. rhabdophora* has simply never evolved a lateral bias. We know that laterality in some species is heritable [33]. Further, we know that lateralized behavior exists in the closely related species, *Brachyrhaphis episcopi* [22] and in several other species in the family Poeciliidae, including, *Xenophallus umbratilis* [16], *Gambusia holbrooki* [14], and *Gambusia hubbsi* [20]. As more studies of laterality in poeciliid fishes accumulate, it may at some point be possible to explore whether laterality is an ancestral trait that has been lost in some species, or if it has evolved *de novo* in several species independently. Previous work in the fish family

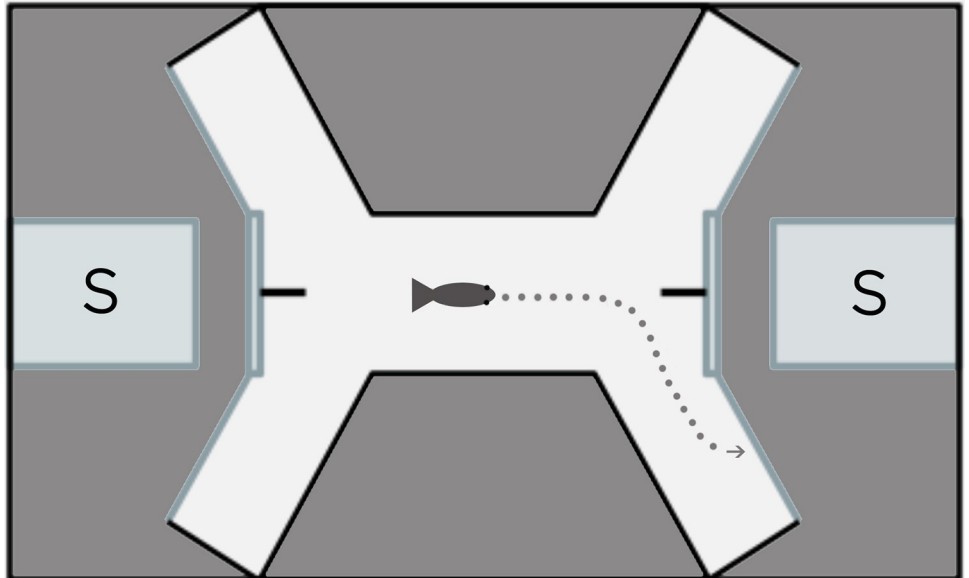

**Fig 3. Mean laterality index scores (+- 2SE) across stimuli for the predator-free (Grande) and predator (Javilla) populations.**

Belontidae showed that the degree of individual lateralization among species occurred independent of the phylogenetic history of the species examined [34], suggesting that in that family at least some aspects of lateralization are due to evolution within individual species.

Another possible explanation is that laterality in *B. rhabophora* has been lost or has never evolved because it is evolutionarily costly. We know that in general there can be benefits and costs of laterality [35]. For example, some lateralized fish are twice as fast at catching prey as non-lateralized fish when a predator is present [36] and schools of lateralized fish perform more cohesively and coordinated than schools of non-lateralized fish [37]. However, there is also evidence that non-lateralized individuals tend to choose higher-quality shoals (defined as those that are larger in size and containing members of the same size as the individual) than lateralized individuals [38]. Wiper [1] suggested several potential costs of laterality, including the possibility that predators could exploit lateralized predator-avoidance behaviors in prey species. Hence, it is possible that lateralized behavior is simply too costly in *B. rhabdophora* when predator encounters are frequent. That is, if an individual prefers one eye to view a predator when predators are common, this could increase overall prey mortality risk. We know that *B. rhabdophora* initially evolved in the presence of predators, although several populations have migrated to environments free of piscivory [23]. Hence, in our system this pattern of colonization could explain why neither the predator nor predator-free populations displayed lateralized detour behavior.

## How common is laterality?

Our failure to detect lateralized behavior is uncommon in published research (but see [39]). Yet, it is still not clear how ubiquitous laterality is in nature, and whether once it is gained is it likely to persist or to be lost. There is likely a publication bias favoring reports of species that show lateralization. Most work that examines the costs of laterality uses species with some element of laterality present [19, 40–42]. What is missing are studies that examine the costs of laterality in bilaterally symmetrical species without laterality, including species with ancestors that possessed the trait. *Brachyrhaphis rhabdophora* might be a good model for this type of work. This species occurs in both predator and predator-free environments and yet still does not display laterality. Whether or not this is associated with potential costs imposed by predation is unclear. More work is needed to understand if predation really plays a role in influencing lateralized behavior in this system.

## Supporting information

**S1 Data.**
(CSV)

## Acknowledgments

This research was conducted under the IACUC protocol 18–0803. We thank Javier Guevara Siquiera and Lourdes Vargas Fellas at the Vide Sylvestre, Ministrio del Ambiente y Energia (MINAE), Sistema Nacional de Áreas de Conservación (SINAC), Costa Rica, for processing our collecting permits. Additionally, we appreciate Alli Duffy, Becca White, Trevor Williams, and Megan Pew for contributing to this project by aiding with fieldwork, specimen processing, and fish care. Culum Brown and two anonymous reviewers provided constructive feedback during the review process.

## Author Contributions

**Conceptualization:** Maren G. Callaway, Erik S. Johnson, Jerald B. Johnson.

**Data curation:** Maren G. Callaway, Erik S. Johnson.

**Formal analysis:** Maren G. Callaway, Erik S. Johnson.

**Funding acquisition:** Maren G. Callaway, Jerald B. Johnson.

**Investigation:** Maren G. Callaway, Erik S. Johnson, Jerald B. Johnson.

**Methodology:** Maren G. Callaway, Erik S. Johnson, Jerald B. Johnson.

**Project administration:** Maren G. Callaway, Jerald B. Johnson.

**Resources:** Jerald B. Johnson.

**Supervision:** Erik S. Johnson, Jerald B. Johnson.

**Validation:** Erik S. Johnson, Jerald B. Johnson.

**Visualization:** Maren G. Callaway, Erik S. Johnson.

**Writing – original draft:** Maren G. Callaway, Jerald B. Johnson.

**Writing – review & editing:** Maren G. Callaway, Erik S. Johnson, Jerald B. Johnson.

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
