## [Decision Letter · Decision Letter 0]

19 May 2022

PONE-D-22-06171Predation history has no effect on lateralized behavior in *Brachyrhaphis rhabdophora*PLOS ONE

Dear Dr. Callaway,

Thank you for submitting your manuscript to PLOS ONE. After careful consideration, we feel that it has merit but does not fully meet PLOS ONE’s publication criteria as it currently stands. Therefore, we invite you to submit a revised version of the manuscript that addresses the points raised during the review process. Please submit your revised manuscript by Jul 03 2022 11:59PM. If you will need more time than this to complete your revisions, please reply to this message or contact the journal office at plosone@plos.org. Please include the following items when submitting your revised manuscript:A rebuttal letter that responds to each point raised by the academic editor and reviewer(s). You should upload this letter as a separate file labeled 'Response to Reviewers'.A marked-up copy of your manuscript that highlights changes made to the original version. You should upload this as a separate file labeled 'Revised Manuscript with Track Changes'.An unmarked version of your revised paper without tracked changes. You should upload this as a separate file labeled 'Manuscript'.

We look forward to receiving your revised manuscript.

Kind regards,

Lesley Joy Rogers, B.Sc. (Hons), D.Phil., D.Sc.

Academic Editor

PLOS ONE

Journal Requirements:

2. n your Methods section, please include a comment about the state of the animals following this research. Were they euthanized or housed for use in further research? If any animals were sacrificed by the authors, please include the method of euthanasia and describe any efforts that were undertaken to reduce animal suffering.

[This research was conducted under the IACUC protocol 18-0803. This work was funded by a CURA scholarships from the BYU College of Life Sciences to MC and ESJ. We thank Javier Guevara Siquiera and Lourdes Vargas Fellas at the Vide Sylvestre, Ministrio del Ambiente y Energia (MINAE), Sistema Nacional de Áreas de Conservación (SINAC), Costa Rica, for processing our collecting permits. Additionally, we appreciate Alli Duffy, Becca White, Trevor Williams, and Megan Pew for contributing to this project by aiding with fieldwork, specimen processing, and fish care.]

 [Life Sciences College Underground Research Award (CURA) - MC

- Amount awarded: $1500

- URl: https://lscura.byu.edu/]

6. We note that Figure 1 in your submission contain map/satellite images which may be copyrighted. All PLOS content is published under the Creative Commons Attribution License (CC BY 4.0), which means that the manuscript, images, and Supporting Information files will be freely available online, and any third party is permitted to access, download, copy, distribute, and use these materials in any way, even commercially, with proper attribution. For these reasons, we cannot publish previously copyrighted maps or satellite images created using proprietary data, such as Google software (Google Maps, Street View, and Earth). For more information, see our copyright guidelines: http://journals.plos.org/plosone/s/licenses-and-copyright.

a) You may seek permission from the original copyright holder of Figure 1 to publish the content specifically under the CC BY 4.0 license.  

Additional Editor Comments:

The reviewers ask for major revision. Please take all of their comments into account before you re-submit a much revised copy.

Reviewers' comments:

Reviewer's Responses to Questions

**Comments to the Author**

1. Is the manuscript technically sound, and do the data support the conclusions?

Reviewer #1: No

Reviewer #2: Partly

2. Has the statistical analysis been performed appropriately and rigorously? 

Reviewer #1: No

Reviewer #2: Yes

3. Have the authors made all data underlying the findings in their manuscript fully available?

Reviewer #1: No

Reviewer #2: Yes

4. Is the manuscript presented in an intelligible fashion and written in standard English?

Reviewer #1: Yes

Reviewer #2: Yes

5. Review Comments to the Author

Reviewer #1: This paper set out to determine the impact of predation pressure on laterality in a poeciliid in Costa Rica. Unfortunately, with just 2 rivers examined (1 high and 1 low predation pressure) it can not examine the hypothesis as stated. The data about the response to the various stimuli may be useful but it is difficult to know how (if) they should be combined owing to the variation in populations from which they were drawn. The authors also make several statements attempting to enhance the apparent novelty of this study, but seem to ignore the literally 100s of papers that have addressed this topic previously. Data analysis is limited.

Specific comments.

L35-36: Rephrase this. There must be 100 papers on laterality in fishes. You also contradict this statement L152 as the opener to the Discussion.

L46: Odd that till now there is no mention of the work on Brachyrphaphis episcopi. A sister species where a number of papers have examined high and low predation effects on laterality.

L61: Im not so sure about these hypotheses. If predation drives the evolution and or development of laterality then shouldn’t the high predation population be highly lateralised irrespective of the stimuli. Otherwise you are suggesting domain specific laterality which would be rather unusual. Also these 2 hypotheses are clearly not independent of one another. What we are expecting is a population by stimuli interaction of some sort.

L64: I would delete the last line. This is the introduction not the results section.

L75: with just one population from each of high and low predation the authors can not generalise about the potential impact of predation pressure on laterality. There are presumably any number of environmental variables that differ between these two streams that could result in variation in laterality (or not).

L118: did a male have to make a certain number of choices to be included in the data set? Eg more than 10?

119: its important to describe the lighting above the tank and discrepancies in shadow can induce false positives.

L139: Im not sure why there was no attempt to use a 2 way-anova to examine laterality scores. Nor is there any mention of absolute laterality which tends to be a good predictor of the strength of laterality irrespective of direction.

Fig 1? My version doesn’t seem to have a map.

Fig 3. Why 2 SE ?

L161-167: This explanation does not make sense. Clearly populations within a species vary with respect to predation regime, so to suggest that a sister species ought to show the same pattern due to a common ancestor is irrational. Moreover, Bisazza and colleagues have conducted multiple comparisons across species and found that schooling tendency predicted laterality not phylogeny.

L168: also not true. There are multiple papers examining the costs and benefits of laterality in poeciliids as well as in vertebrates more generally, including multiple reviews.

L178: There is most certainly a publication bias. I have three students working on laterality in 5 species and is not reliably showing up in any of them.

Reviewer #2: Glad to read this paper that aimed at investigating behavioural lateralisation in a poeciliid fish depending on predation history. Although well written and organized my opinion is that there are some flawless in the current form of the paper. First of all it is not clear to me how many trials were recorded in order to compute the index of laterality for each subject during the 20min of observation, whether or not they have discarded subjects that does not spontaneously turn or if they tend make "proper turn" toward one or the other end of the apparatus. I was also wondering about the possibility to test for possible behavioural asymmetries using a viewing situation in which the subject is required to make closer inspection to the stimuli. Finally why do not include females?

In the discussion section (line 167) the authors claim that there are "too few tests of laterality....", not sure what they are arguing here but actually theres a quite large literature on how to assess lateralization in poeciliid fish.

In general the result section appear to be pretty much concise.

Minor comments:

reference 31: Giorgio V. should be Vallortigara G

6. PLOS authors have the option to publish the peer review history of their article (what does this mean?). If published, this will include your full peer review and any attached files.

Reviewer #1: No

Reviewer #2: No

---

## [Author Response · Author response to Decision Letter 0]

5 Jul 2022

Our response has been included in the "response to reviewers" file which I have attached.

---

## [Decision Letter · Decision Letter 1]

7 Sep 2022

PONE-D-22-06171R1Predation history has no effect on lateralized behavior in *Brachyrhaphis rhabdophora*PLOS ONE

Dear Dr. Callaway,

Thank you for submitting your manuscript to PLOS ONE. After careful consideration, we feel that it has merit but does not fully meet PLOS ONE’s publication criteria as it currently stands. Therefore, we invite you to submit a revised version of the manuscript that addresses the points raised during the review process.

The manuscript has improved a great deal. The reviewers suggest you check on a handfull of things and clarify certain points. Most important of those are the following.

Figure 3 shows that there is a very large variation in the data, and we must consider the causes of this variation. Follow rev 3 and explore this further with graphs and statistics.Rev 1 is still uncomfortable with the one population from high and one from low predation environments (n = 1). These populations undoubtedly differ in any number of ways which may influence laterality. Adjust interpretation accordingly.Several issues on methodology and statistics have to be clarified. including how long males recovered before being tested again?Are there variation by pop and treatment – analyze with a mixed model with individual ID as the random variable, pop and treatment as independent variables and LI as the dependent?How was the variation around means? Where any of the individuals lateralised in their response to various cues? Eg. Where strongly lateralised fish consistently strongly lateralised?Provide the raw data as csv file, with better annotation of columns and summaries.

We look forward to receiving your revised manuscript.

Kind regards,

Arnar Palsson, Ph.D.

Academic Editor

PLOS ONE

Journal Requirements:

Additional Editor Comments (if provided):

The manuscript has improved a great deal. The reviewers suggest you check on a handfull of things and clarify certain points. Most important of those are the following.

1. Figure 3 shows that there is a very large variation in the data, and we must consider the causes of this variation. Follow rev 3 and explore this further with graphs and statistics.

2. Rev 1 is still uncomfortable with the one population from high and one from low predation environments (n = 1). These populations undoubtedly differ in any number of ways which may influence laterality. Adjust interpretation accordingly.

3. Several issues on methodology and statistics have to be clarified. including

a. how long males recovered before being tested again?

b. Are there variation by pop and treatment – analyze with a mixed model with individual ID as the random variable, pop and treatment as independent variables and LI as the dependent?

c. How was the variation around means? Where any of the individuals lateralised in their response to various cues? Eg. Where strongly lateralised fish consistently strongly lateralised?

4. Provide the raw data as csv file, with better annotation of columns and summaries.

Reviewers' comments:

Reviewer's Responses to Questions

**Comments to the Author**

1. If the authors have adequately addressed your comments raised in a previous round of review and you feel that this manuscript is now acceptable for publication, you may indicate that here to bypass the “Comments to the Author” section, enter your conflict of interest statement in the “Confidential to Editor” section, and submit your "Accept" recommendation.

Reviewer #1: (No Response)

Reviewer #3: All comments have been addressed

2. Is the manuscript technically sound, and do the data support the conclusions?

Reviewer #1: Partly

Reviewer #3: Partly

3. Has the statistical analysis been performed appropriately and rigorously? 

Reviewer #1: No

Reviewer #3: Yes

4. Have the authors made all data underlying the findings in their manuscript fully available?

Reviewer #1: Yes

Reviewer #3: Yes

5. Is the manuscript presented in an intelligible fashion and written in standard English?

Reviewer #1: Yes

Reviewer #3: Yes

6. Review Comments to the Author

Reviewer #1: PONE-D-22-06171R1

Predation history has no effect on lateralized behavior in Brachyrhaphis rhabdophora

Callaway

This paper has certainly improved and I thank the authors for taking on and addressing previous concerns. I confess, even given previous work on this system, I’m still uncomfortable with n = 1.

Having done many of these projects myself, I recognize how much work goes into it, and I want to be sure the authors have really explored their data as best they can. With this in mind I’ve offered some alternative analyses that they might consider.

The methods still require a bit more detail so folks could replicate them.

Specific comments:

L65 I still don’t feel comfortable with the one population from high and one from low predation environments. These populations undoubtedly differ in any number of ways which may influence laterality. However, given the historical context (Johnson and Belk 2001) it is reasonably likely that the variation may due to predation. I would be far happier to see multiple pops tested.

L126: Its also not clear how long males recovered before being tested again. (inter-trial period). Also state here that you did not set a min number of turns for the males to qualify and perhaps make a statement about how many runs the fish did make.

L144: Im still a bit confused by the statistics. I can appreciate that the current method tells us if the estimates of LI overlap with 0, but I still think its worth-while exploring if they differ with pop and treatment as this speaks directly to your hypotheses. In this case, would it not be better to include a mixed model with individual ID as the random variable, pop and treatment as independent variables and LI as the dependent?

L157: I would be interested to know if any of the individuals were lateralised in their response to various cues. What does the distribution look like? Where strongly lateralised fish consistently strongly lateralised? Previous papers, for instance, have examined the proportion of the population tested that scored at 70% L or R (lateralised). I just get the feeling you could do more with this data and individual details get lost in population averages.

L203: of course there are other explanations, including low sample sizes or inappropriate experimental design, which are less interesting but nonetheless significant concerns.

Reviewer #3: The paper “Predation history has no effect on lateralized behavior in Brachyrhaphis

rhabdophora”

The paper of Callaway and coll. investigate the effect of predation history on eye preference in the livebearing fish Brachyrhaphis rhabdophora by detour test. The authors reported no evidence for behavioral lateralization at all in response to any stimuli (predator, mate, novel, control), regardless of predation population or predation-free population.

This paper is written clearly and the unique behavioral experiment has been conducted. Appropriate corrections have already been noted by the reviewers. I feel that this manuscript provides a novel contribution to the field. Interesting concept and idea but there is a lot of variation in the behavioral data and no explanation for this. Also, I point out that there are problems with the experimental design.

Please make a couple of improvements as follows:

Introduction

P4, line48-53: It is necessary to explain in detail how much mortality in the fish differs between environments with and without predators.

Methods

P7, line127-129: Insufficient information about the experimental procedure: in what order were the four stimuli presented? Were the stimuli random order? The interval of the experiment was short (10 minutes), and it is suspected that the subjects may become accustomed to the experimental test and feel tired.

Result

Figure 3 shows that there is a very large variation in the data, and we must consider the causes of this variation. This is because this result is directly related to the conclusion that there is no laterality in the livebearing fish. One possibility is that a detour test of 20 minutes may be too long for observation. Even at the beginning and end of the experiment, the laterality tendency eye may change (Cantalupo et al. Neuropsychologia 1995, Sovrano et al. Physiology and Behavior 2001, Bisazza et al. Behavioural Brain Research 2002). If LI is calculated for only the first 5 or 10 minutes, will the results change? Longer experiment time introduces the effect of habituation and learning, which makes the interpretation of results more difficult.

The number of trials per individual is not specified in the results. It is unfortunate that this has not been corrected in the text, even though this was pointed out by reviewer 2.

Discussion

P9, line167-171: It is necessary to specify what type of left-right behavior is exhibited by the three closely related species of this species. Whether there is a unifying trend or not would allow us to consider the need to question the phylogenetic background.

P10, line181-183: "non-lateralized individuals tend to choose higher-quality shoals than lateralized individuals"

What is meant by “higher-quality shoals” and how is it costly needs to be explained.

7. PLOS authors have the option to publish the peer review history of their article (what does this mean?). If published, this will include your full peer review and any attached files.

Reviewer #1: **Yes: **Prof. Culum Brown

Reviewer #3: No

---

## [Author Response · Author response to Decision Letter 1]

9 Jan 2023

22 October 2022 

Dear Editor,

Thank you for the additional input on our manuscript titled “Predation history has no effect on lateralized behavior in Brachyrhaphis rhabdophora” (PONE-D-22-06171R1). We have carefully read the comments made by you and by the two reviewers, and we have revised our manuscript accordingly. Here we provide an itemized list of each point raised by the reviewers (listed in bold text), along with our response (listed in non-bold text). 

We appreciate this additional input. It has furthered improved the paper, and we hope that you will find it acceptable for publication. Please contact me if there are any further edits that you would like us to make. We look forward to hearing from you.

Sincerely,

Maren Callaway

REVIEWER #1 – Professor Culum Brown

This paper has certainly improved, and I thank the authors for taking on and addressing previous concerns. I confess, even given previous work on this system, I’m still uncomfortable with n = 1. Having done many of these projects myself, I recognize how much work goes into it, and I want to be sure the authors have really explored their data as best they can. With this in mind, I’ve offered some alternative analyses that they might consider. The methods still require a bit more detail so folks could replicate them.

1. Line 65 - I still don’t feel comfortable with the one population from high and one from low predation environments. These populations undoubtedly differ in any number of ways which may influence laterality. However, given the historical context (Johnson and Belk 2001) it is reasonably likely that the variation may be due to predation. I would be far happier to see multiple pops tested.

We appreciate this comment. Unfortunately, we do not have replicate populations in the laboratory to conduct additional tests. However, we have done a couple of things in the revised manuscript that we hope will acknowledge the concern and still allow us draw conclusions from the study. First, we have added language to the revised manuscript that makes clear that when we refer to ‘predation environment’ in the paper, that we include not only the presence or absence of predators, but also other ecological components associated with these environment types. Specifically, we refer to Johnson (2002) which included a very transparent description of these habitat types. This should make it clear to the reader that predator environments include several factors, which is certainly the case for previous work done in B. episcopi and guppies. It’s not an ideal response, but it does acknowledge previous work in similar systems. Second, we also refer to published work (Johnson and Zuniga-Vega 2009) showing that these two specific populations are known to differ in mortality rates. We don’t know exactly all the factors that contribute to these differences—although the cichlid fish predator must be an important part of this—but we do know that for the two localities evaluated here, there are measured differences. Combined, we hope this will at least add some confidence to the comparison between the two sites chosen. It’s not perfect, but it is also not as fraught with confounders in other systems where less is known.

2. L126: It’s also not clear how long males recovered before being tested again. (inter-trial period). Also state here that you did not set a minimum number of turns for the males to qualify and perhaps make a statement about how many runs the fish did make.

Thank you for asking for this clarification. We agree, it is important to make sure that there are not some trials with very few turns that that are unduly influencing our results. Consequently, we re-ran our analyses including only trials where fish made at least five turns—these results also showed no differences among populations. In essence, nothing changed. That said, we did clarify this in our revised manuscript in the Methods section (line 111). We also now report how long the resting period was between trials (line 124).

3. L144: I’m still a bit confused by the statistics. I can appreciate that the current method tells us if the estimates of LI overlap with 0, but I still think its worth-while exploring if they differ with pop and treatment as this speaks directly to your hypotheses. In this case, would it not be better to include a mixed model with individual ID as the random variable, pop and treatment as independent variables and LI as the dependent?

This is a sound suggestion, especially as it allows us to compare our results directly to the work previously done with B. episcopi. As recommended, we ran a mixed model repeated measures ANOVA (with all samples included), treating population and treatment as independent variables, and individual as a random variable (see lines 153-155). Consistent with our previous analyses of LI for each treatment, we found no differences between predator environments, and no differences among our treatments. 

4. L157: I would be interested to know if any of the individuals were lateralised in their response to various cues. What does the distribution look like? Where strongly lateralised fish consistently strongly lateralised? Previous papers, for instance, have examined the proportion of the population tested that scored at 70% L or R (lateralised). I just get the feeling you could do more with this data and individual details get lost in population averages.

Thanks for this suggestion. We went back to our data and found that none of the individuals in our study had LI values that exceeded positive or negative 0.7. We report this in our Results (line 162).

5. L203: Of course, there are other explanations, including low sample sizes or inappropriate experimental design, which are less interesting but nonetheless significant concerns.

Our sample size and methodological approach follow typical conventions for this type of study. Using a similar approach, Johnson et al. (2020) found evidence for lateralized behavior in Xenophallus umbratilis, and Brown et al (2004) found evidence for lateralized behavior in B. episcopi from predator environments. Hence, we feel reasonably confident that our findings are real, and not just due to design issues.

REVIEWER #3

The paper of Callaway and colleagues investigates the effect of predation history on eye preference in the livebearing fish Brachyrhaphis rhabdophora by detour test. The authors reported no evidence for behavioral lateralization at all in response to any stimuli (predator, mate, novel, control), regardless of predation population or predation-free population.

This paper is written clearly and the unique behavioral experiment has been conducted. Appropriate corrections have already been noted by the reviewers. I feel that this manuscript provides a novel contribution to the field. Interesting concept and idea but there is a lot of variation in the behavioral data and no explanation for this. Also, I point out that there are problems with the experimental design. Please make a couple of improvements as follows:

Introduction

6. P4, line 48-53: It is necessary to explain in detail how much mortality in the fish differs between environments with and without predators.

Thank you for this request. One of the strengths of this system is that we know mortality rates by size class for B. rhabdophora from these two environments. This work is published in Johnson and Zuniga-Vega 2009. We have included a brief reference to these mortality rates in our Methods section where we describe the study system. 

7. Methods. P7, line127-129: Insufficient information about the experimental procedure: in what order were the four stimuli presented? Were the stimuli random order? The interval of the experiment was short (10 minutes), and it is suspected that the subjects may become accustomed to the experimental test and feel tired.

Thank you, these are all important questions that we have now addressed in the methods of our paper. In brief, we ran the control treatment first and the predator treatment last for each species, with the other two treatments in between. We have added an explanation for this order at line 122 of the revised paper. As mentioned above, we explored the possibility that fish might have become accustomed to the test and therefore decreased their activity and detour behavior. We addressed this by examining if fish conducted fewer detours in the latter half of a trial; we found no evidence for this.

8. Results. Figure 3 shows that there is a very large variation in the data, and we must consider the causes of this variation. This is because this result is directly related to the conclusion that there is no laterality in the livebearing fish. One possibility is that a detour test of 20 minutes may be too long for observation. Even at the beginning and end of the experiment, the laterality tendency eye may change (Cantalupo et al. Neuropsychologia 1995, Sovrano et al. Physiology and Behavior 2001, Bisazza et al. Behavioural Brain Research 2002). If LI is calculated for only the first 5 or 10 minutes, will the results change? Longer experiment time introduces the effect of habituation and learning, which makes the interpretation of results more difficult.

This was a good suggestion to evaluate. We re-ran our analyses including only the first 10 minutes of data in each trial. We found no difference in our results. Habituation does not appear to be an issue here.

9. The number of trials per individual is not specified in the results. It is unfortunate that this has not been corrected in the text, even though this was pointed out by Reviewer 2.

As stated on line 122, each individual participated in four trials, one for each stimulus.

10. Discussion. P9, line167-171: It is necessary to specify what type of left-right behavior is exhibited by the three closely related species of this species. Whether there is a unifying trend or not would allow us to consider the need to question the phylogenetic background.

Of the four species that we refer to here, three are distantly related and one is closely related. We do not have information on behavioral lateralization in all species in the family. What we know is this. The closely-related species B. episcopi shows behavioral lateralization in individuals that occur with predators, but not in individuals that occur with no predators. The other three species are from distantly related genera. Hence, it seems unlikely that we are dealing with an issue of phylogenetic history governing laterality patterns. Though there is evidence of heritability of laterality in some species (line 178), we know laterality and phylogenetic history are independent of each other in another fish family, Belontidae (line 184). As such, we question the phylogenetic background given there is evidence for both possibilities. 

11. P10, line181-183: "non-lateralized individuals tend to choose higher-quality shoals than lateralized individuals". What is meant by “higher-quality shoals” and how it is costly needs to be explained.

Thank you for asking for this clarification. We have added a phrase in this sentence (line 193) that defines what we mean by high-quality shoals: shoals with more individuals and shoals with individuals of roughly the same size as the fish present. By costly, we simply refer to the idea that the larger the shoal the greater the dilution for each individual. We also note that individuals of the same size are less conspicuous than individuals of different sizes.

---

## [Editor Report · Decision Letter 2]

11 Jan 2023

Predation history has no effect on lateralized behavior in *Brachyrhaphis rhabdophora*

PONE-D-22-06171R2

Dear Dr. Callaway,

We’re pleased to inform you that your manuscript has been judged scientifically suitable for publication and will be formally accepted for publication once it meets all outstanding technical requirements.

Kind regards,

Arnar Palsson, Ph.D.

Academic Editor

PLOS ONE
---

## [Editor Report · Acceptance letter]

6 Feb 2023

PONE-D-22-06171R2 

Predation history has no effect on lateralized behavior
in *Brachyrhaphis rhabdophora*

Dear Dr. CALLAWAY:

I'm pleased to inform you that your manuscript has been deemed suitable for publication in PLOS ONE. Congratulations! Your manuscript is now with our production department. 

Kind regards, 

on behalf of

Dr. Arnar Palsson 

Academic Editor

PLOS ONE